

# Physical Protection of Soil Carbon Stocks Under Regenerative Agriculture

**Sam G. Keenor [1], Rebekah Lee[1] and Brian J. Reid[1*]**

[1]School of Environmental Sciences, University of East Anglia, Norwich, NR4 7TJ, UK.

*Corresponding author: e-mail: b.reid@uea.ac.uk Tel: +44(0)1603 592357
ORCID-ID: Sam G. Keenor https://orcid.org/0000-0001-8620-8133
ORCID-ID: Brian J. Reid https://orcid.org/0000-0002-9613-979X

**Keywords:** Soil Carbon, Carbon Protection, Stabilised Carbon, Recalcitrant Carbon, Occluded Carbon, Regenerative Agriculture, Aggregate Stability.

**Abstract**

Regenerative agriculture is emerging as a strategy for carbon sequestration and climate change mitigation. However, for sequestration efforts to be successful, long-term stabilisation of Soil Organic Carbon (SOC) is needed. This can be achieved either through the uplift in recalcitrant carbon stocks, and/or through physical protection and occlusion of carbon within stable soil aggregates. In this research, soils from blackcurrant fields under regenerative management (0 to 7 years) were analysed with respect to: soil bulk density (SBD), aggregate fractionation (water stable aggregates vs. non-water stable aggregates (WSA and NWSA respectively)), soil carbon content, and carbon stability (recalcitrant vs. labile carbon). From this, long term carbon sequestration potential was calculated from both recalcitrant and physically occluded carbon stocks (stabilised carbon). Results indicated favourable shifts in the proportion of NWSA:WSA with time. This ratio increasing from 27.6% : 5.8% (control soil) to 12.6% : 16.0% (alley soil), and 16.1% : 14.4% (bush soil) after 7 years. While no significant ($p \geq 0.05$)) changes in recalcitrant carbon stocks were observed after 7 years, labile carbon stocks increased significantly ($p \leq 0.05$) from 10.44 t C ha$^{-1}$ to 13.87 t C ha$^{-1}$. As a result, total sequesterable carbon (*stabilised carbon*) increased by 1.7 t C ha$^{-1}$



over the 7 year period, due to the occlusion and protection of this labile carbon stock within
WSA fraction. This research provides valuable insights into the mechanisms of soil carbon
stabilisation under regenerative agriculture practices and highlights the importance of soil
aggregates in physically protecting carbon net-gains.
**1. Introduction**
Land use change, conventional land management practice, and aggressive agricultural
techniques remain key drivers of soil damage and degradation (Lal, 2001, Lambin et al., 2001,
Foley et al., 2005, Pearson, 2007, Smith, 2008, Al-Kaisi and Lal, 2020). Without a shift to more
sustainable approaches future agricultural productivity will be endangered, and with it the
loss of food and economic security for many around the world (Zika and Erb, 2009, Tilman et
al., 2011, Sundström et al., 2014).
The effects of soil degradation can greatly reduce environmental and ecosystem quality
and function (IPBES, 2018). Soil erosion and loss of soil organic carbon (SOC), structural
damage (destruction of soil aggregates and compaction), contamination, salinisation, and
nutrient depletion all contribute to soil degradation (Lal, 2015, Montanarella et al., 2016,
Sanderman et al., 2017); undermining  the provision of key ecosystem services that underpin
wider environmental health and function (Dominati et al., 2010, Power, 2010).
At landscape scales, soil degradation compounds and threatens desertification and
biodiversity loss (Zika and Erb, 2009, Power, 2010, Orgiazzi and Panagos, 2018, Huang et al.,
2020), while making significant contributions to greenhouse gas emissions and climate
change (Lal, 2004, Smith et al., 2020). Globally, agriculture is associated with roughly a third
of total land use and nearly a quarter of all global greenhouse gas emissions each year (Foley
et al., 2011, Smith et al., 2014, Newton et al., 2020). To date it is estimated that more than
176 Gt of soil carbon has been lost to the atmosphere (IPBES, 2018), with approximately 70-



80% of this *(~130 - 140 Gt)* as a direct consequence of anthropogenic land management and
soil cultivation (Sanderman et al., 2017, Lal et al., 2018, Smith et al., 2020). Meanwhile the
area of land affected by desertification globally has been reported to  exceed  25% and is
expanding each year (Huang et al., 2020).

A key mechanistic step in the degradation of soil, is the loss and destruction of stable soil

aggregates and loss of SOC (Smith, 2008, Baveye et al., 2020). Soil aggregate formation, as
facilitated by SOC, assists the stabilisation and storage of carbon and imparts resilience to
soils against erosion and climate change while providing hydrological benefits and enhancing
soil fertility (Lal, 1997, Abiven et al., 2009, Chaplot and Cooper, 2015, Veenstra et al., 2021).

In addition to mitigating the negative effects of soil degradation, the formation and

persistence of stable soil aggregates is instrumental in soil carbon sequestration (Lal, 1997,
Six et al., 1998, Abiven et al., 2009). Particularly due to physical protection of labile carbon
within the soil aggregates which minimise biogenic and oxidative decay of SOC (Brodowski et
al., 2006, Smith, 2008, Schmidt et al., 2011, Berhe and Kleber, 2013).

However, it is important, when viewed through the lens of carbon sequestration that we

acknowledge not all carbon is equal. The potential for long-term carbon sequestration is
governed by the resistance of the carbon to degradation. This resistance being conferred
through i) inherent recalcitrance of the carbon, and ii) physical protection of the carbon and
occlusion within soil aggregates. Thus, when considering carbon sequestration potentials as
solutions to climate change it is imperative that we differentiate between soil carbon which
is transient and soil carbon which endures.

By adopting of more sustainable management practices, agriculture can transition from a

negative to a positive force for the environment; providing and enhancing a variety of key
ecosystem services (*water regulation, soil property regulation, carbon sequestration and*



*biodiversity support* (de Groot et al., 2002, Dominati et al., 2010, Power, 2010, Baveye et al.,
2016, Keenor et al., 2021)).

Regenerative agriculture offers opportunities to produce food and other agricultural

products with minimal negative, or even net positive outcomes for society and the
environment; potentially improving farm profitability, increasing food security and resilience,
and helping to mitigate climate change (Al-Kaisi and Lal, 2020, Newton et al., 2020). Despite
having no single definition or prescriptive set of criteria, regenerative agriculture is widely
understood to include the key concepts of: (i) reducing/limiting soil disturbance; (ii)
maintaining continuous soil cover (as vegetation, litter or mulches), (iii) increasing quantities
of organic matter returned to the soil; (iv) maximising nutrient and water-use efficiency in
crops; (v) integrating livestock; (vi) reducing or eliminating synthetic inputs (fertilisers and
pesticides); and (vii) increasing and broadening stakeholder engagement and employment
(Newton et al., 2020, Paustian et al., 2020, Giller et al., 2021).

Adoption of no/minimum-till techniques increases the extent of soil aggregation and

improves long-term carbon storage potential (Lal, 1997, Gál et al., 2007, Ogle et al., 2012,
Lehmann et al., 2020). Furthermore, in addition to providing physical protection to more
labile forms of soil carbon, improved soil aggregation enhances resilience to the effects of
drought and erosion, and provides better hydrological function and structure to the soil
(Abiven et al., 2009, Bhogal et al., 2009, Baveye et al., 2020, Ferreira et al., 2020, Martin and
Sprunger, 2022). No/minimum till techniques have been adopted worldwide and in a variety
of agricultural contexts to help reduce soil erosion, increase crop yields  and minimise input
costs all while building soil organic matter (Sisti et al., 2004, Pittelkow et al., 2015, Ferreira et
al., 2020). Adoption of minimum-till and no-till methods compared with conventional tillage
has been reported to significantly increase SOC content within the top 30cm of a soil (Gál et



al., 2007, Ogle et al., 2012). However, these potential SOC increases depend on agricultural
context, climate and soil type (Lal, 2004). Conversion from conventional to regenerative
approaches may increase macro-aggregation and aggregate stability (Lal, 1997), and by
extension, provide the means to protect labile soil carbon; thus, enhancing long-term soil
carbon sequestration efforts (Six et al., 1998, Brodowski et al., 2006, Smith, 2008, Schmidt et
al., 2011, Berhe and Kleber, 2013). Furthermore, adoption of regenerative methods such as
no-till or reduced till can also lessen machinery costs, working hours and direct carbon
emission (Kasper et al., 2009). Indeed, resulting from the adoption of no-till methods, it is
estimated that emission reductions of approximately 241 Tg $CO_2$e have been achieved
globally since the 1970s (Al-Kaisi and Lal, 2020).
To evaluate the influence of transitioning to soft fruit production under regenerative
principles, from a regime of conventional cropping and tillage, a field experiment was
undertaken on a commercial blackcurrant farm in Norfolk, UK. The experiment evaluated 5
blackcurrant fields managed under regenerative principles for increasing lengths of time, and
a conventionally managed arable field evaluated as a datum. The research assessed carbon
stocks across the regimes and thereafter the proportion of carbon stocks associated with the
soil fractions: sand, water stable aggregates (WSA) and non-water stable aggregates (NWSA).
Thermogravimetric Analysis (TGA) was used to differentiate labile and recalcitrant carbon
pools, and their association to the respective soil fractions (Mao et al., 2022). The research
sought to test the hypothesis that a switch from conventional arable farming to regenerative
soft fruit production would increase total soil carbon stock with time and that this carbon
stock would become increasingly stabilised, either associated with WSA (i.e. physically
protected) and/or of greater resistance to degradation (i.e. recalcitrant).
**2. Methods**



**2.1 Field experiment**


This research was undertaken at Gorgate Farm, Norfolk, UK (52°41'58"N 0°54'01"E). The
farm is part of the wider Wendling Beck Environment Project (WBNRP, 2024) a regenerative
farming and landscape management program set in C. 750 ha. The field experiment
comprised 5 blackcurrant fields established in 2019, 2017, 2015, and 2013 (1, 3, 5, and 7 years
since soil disturbance) and a conventionally managed arable field as a datum (0 years since
soil disturbance; field history in the arable regime (2014-2021) is shown in (**Fig. SI 1**. in the
supplement).
The blackcurrant fields under regenerative management were planted using a conservation
strip tillage approach, with the blackcurrant bushes planted as field length strips, leaving
alleyways approximately 2m wide. Currants bushes occupied approximately 40% of the field
and the alleyways between the crops approximately 60%. Once planted, the blackcurrant
crop required minimal interventions beyond the yearly harvest, pruning, sowing of cover
crops in the alleys and fertilisation. Fields remained covered year-round between the
blackcurrant crop, with a diverse grazing cover crop through the autumn and winter months,
and a summer fallow covering crop during the spring and summer months, both directly
drilled, and are treated with sprays of compost tea and organic fertiliser. Comparatively the
control comprised a conventionally managed arable field adjacent to the blackcurrant fields,
cultivated yearly and drilled with winter wheat, with stubble re-incorporation. Samples were
collected in late June, immediately prior to the harvest of both crops.

**2.2 Soil sampling**


Soil core samples (0 - 7.5cm; n = 5) were collected from beneath the blackcurrant bushes
and at the centre of the alleyways using a soil Dent corer. Further soil core samples (n = 5)
were randomly collected from a conventionally managed arable field. Soil samples were



sealed and retained in cold storage (≤ 4 ˚C) prior to laboratory analysis. Soil cores were
subsequently oven dried (40 ˚C for 24hrs) and soil bulk density calculated (n = 5).
**2.3 Soil fractionation**
Soil fractionations, namely, Water Stable Aggregates (WSA), Non-Water Stable Aggregates
(NWSA) and sand, were established using a capillary-wetting wet sieving method, adapted
from Seybold and Herrick (2001): Briefly, the previously dried bulk density samples (n = 5)
were dry sieved (2 mm) to remove all debris. Subsequently, 2mm sieved bulk soil (100 g) was
placed on 63μm sieves. Thereafter, soil was slowly wetted with de-ionised water. Once damp,
samples were submerged and oscillated under de-ionised water (manually agitated at 30
oscillations per minute in 1.5cm of water for 5 minutes). Material that passed through the
63μm sieve was collected and dried ($40^0$C for 24 hours) and then weighed, this fraction was
defined as NWSA. The soil retained on the 63μm sieve was further processed using in sodium
hexametaphosphate solution (0.02 M), to disaggregate the WSA aggregates and separate
from the sand fraction. The material remaining on the 63μm sieve was then dried ($40^0$C for
24 hours); and designated as the sand fraction. The WSA fraction (That which passed through
the 63μm sieve) was subsequently established by back calculation (**Eq. 1**):

**Eq.1**          $\% \, WSA = \left( \frac{Bulk \, Soil \, Mass_{dry} - (Sand \, Mass_{dry} + NWSA \, Mass \, _{dry})}{Bulk \, Soil \, Mass_{dry}} \right) \times 100$

**2.4 Total C, and N content by elemental analysis**
Dry bulk soil, and soil fractions, were milled to produce a fine powder and samples (20 mg;
n = 4) packed in 8 × 5 mm tin capsules. An elemental analyser (Exeter CHNS analyser (CE440))
was used to determine elemental abundance of C and N. Instruments were pre-treated within



conditioning samples (acetanilide 1900µg), a blank sample (empty capsule) and an organic
blank sample (benzoic acid 1700µg) prior to sample analysis, and standard reference
materials (acetanilide 1500µg) were run alongside samples (every 6[th] run) for QA/QC (a
precision threshold of ± 1SD of the mean from the standard reference material) (Hemming,
N.D.).
**2.5 Thermogravimetric assessment of SOC stability**
Thermal stability of the SOC in bulk soil, NWSA and sand fractions were assessed using a
Thermo-gravimetric analyser (Mettler Toledo TGA/DSC 1). Samples (n=2) were contained in
70 µl platinum crucibles. Samples were heated, in an inert atmosphere, at a rate of $10^{o}$C min$^{-}$
$^{1}$ from $25^{o}$C to $1000^{0}$C. TGA data was subsequently used to ascribe stable/not-stable carbon
and inorganic carbon content of the bulk soil and soil fractions. Data was split into 3 distinct
phases by temperature range according to organic matter attrition windows as stated in Mao
*et al.* (2022): i) $25^{o}$C – $125^{o}$C (moisture evaporation), ii) $125^{o}$C – $375^{o}$C (labile components)
and, iii) $375^{o}$C – $700^{o}$C (recalcitrant components.
**2.6 Carbon Assessment**
Soil carbon was assessed as total SOC, soil fraction C, total labile/recalcitrant C and
physically protected/unstabilised C. In addition, C was further assessed on a total field carbon
stock basis (in t ha$^{-1}$). To calculate the total field carbon stock in t ha$^{-1}$ (for all carbon
measures), the C content of both the alley and bush soils (or the sum of their relative
fractions) was multiplied by the relevant soil bulk density measure and the depth of sampling
(ca. 7.5cm) and subsequently added together with acknowledgment of their proportion of
the field (60% and 40%, respectively), as set out in (**Eq. 2**):
**Eq.2**        $C\ tha^{-1} = \left(0.6\left(C_{Alley} \times SBD_{Alley} \times Depth\right)\right) + \left(0.4\left(C_{Bush} \times SBD_{Bush} \times Depth\right)\right)$



**2.7 Statistical analysis**

Significant differences between the field sites were determined using *post hoc* tests on one-way ANOVA with Tukey's HSD, data significance set to 95% ($p \leq 0.05$) (ANOVA; IBM SPSS 28). Significant differences between the individual regimes within field sites (alley soil vs. bush soil) were determined using two tailed T-tests, with data significance set at two levels of confidence; 95% ($p \leq 0.05$), and 99% ($p \leq 0.01$) (independent samples T-test; IBM SPSS 28).

**3. Results and Discussion**

**3.1 Bulk Density**

Soil bulk density (SBD) provides insights into soil structures, arrangement of soil particles, and the extent of soil aggregation arising from the influence of physical, chemical, and biological edaphic factors (Al-Shammary et al., 2018). As SBD accounts for the total volume that soils occupy (including the mineral, organic and pore space components), they can act as a key soil condition indicator (Chaudhari et al., 2013, Allen et al., 2011). SBD maintains a close correlation to concentrations of organic matter and carbon within the soil, where soils become depleted in carbon SBD tends to increase, potentially leading to compaction of soil structures (Allen et al., 2011).

Land use management can have significant effect upon the physical condition of soils, and by extension the services provided by soils: management that culminates in soil compaction and structural damage reduces available pore space, greatly limiting the storage and infiltration capabilities of water, the depth to which roots can penetrate, and the movement of soil fauna; subsequently impairing the function and productivity of soils (Byrnes et al., 2018, Pagliai et al., 2004).



Soils may be considered compacted where soil resistance limits or inhibits the movement
of roots through the soil (SBD between 1.4 g cm$^{-3}$ (clay rich soils), and 1.8 g cm$^{-3}$ (sand rich
soils)), where SBD is found to exceed these limits negative effects to the growth and
productivity of crops may be observed (Kaufmann et al., 2010, Shaheb et al., 2021).
SBD was observed to decrease significantly ($p \leq 0.05$) in both the alley soils and bush soils
in all regeneratively managed fields relative to the conventional control (**Fig. 1**). The highest
overall SBD was measured in the control soil (1.75 g cm$^{-3}$) and the lowest SBD in the year 3
bush soil (1.07 g cm$^{-3}$) (**Fig. 1**).
In the alley soils SBD was observed to decrease significantly ($p \leq 0.05$) in all of the
regeneratively managed soils compared to the conventional control (**Fig. 1**). Between the
regeneratively managed soils SBD was observed to decrease (not significantly ($p \geq 0.05$))
successively with each additional year under regenerative management; from 1.35 g cm$^{-3}$ in
the year 1 alley soil, to 1.15 g cm$^{-3}$ in the year 7 alley soil (relative to 1.75 g cm$^3$ in the
conventional control soil) (**Fig. 1**).
In the bush soils SBD was also observed to decrease significantly ($p \leq 0.05$) in all
regeneratively managed soils relative to the conventional control (**Fig. 1**). Between the
regeneratively managed soils SBD was observed to generally decrease with time, however
this was not successive; the greatest decrease in SBD (significant ($p \leq 0.05$)) was observed
between the year 1 and year 3 soils, reducing from 1.32 g cm$^{-3}$ in to 1.07 g cm$^{-3}$ , before
increasing (not significantly ($p \geq 0.05$)) in years 5 and 7 (to 1.18 g cm$^3$ and 1.16 g cm$^3$
respectively)(**Fig. 1**).
When compared pairwise, SBD in the alley soils and the bushes soils were observed to be
broadly similar, with only one pair (*year 3*) showing a significant difference ($p < 0.05$) between
the alley and bush soils, measuring 1.27 g cm$^{-3}$ and 1.07 g cm$^{-3}$ respectively (**Fig. 1**).




None of the soils measured in this investigation were observed to exceed the root limiting
soil density factor of 1.8 g cm$^{-3}$ suggesting no significant detriment to the growth of plants

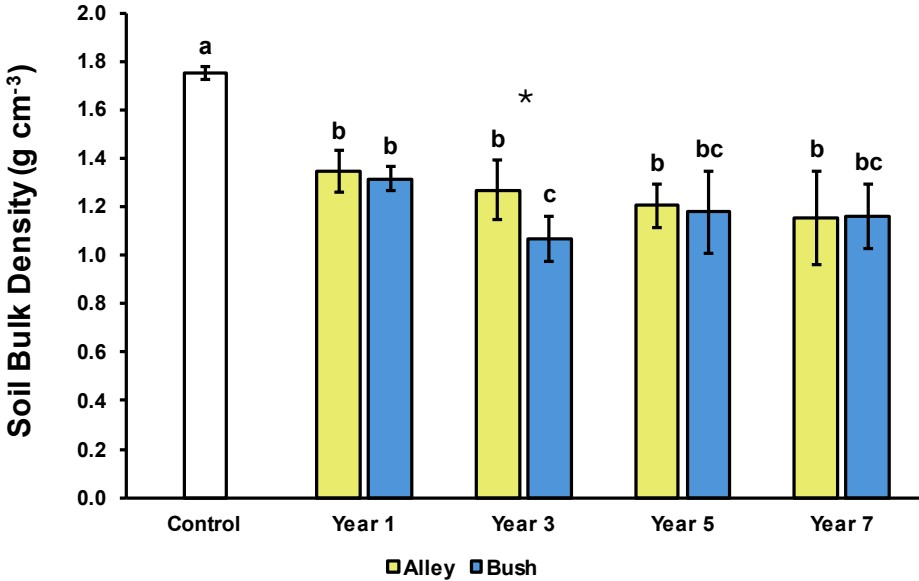

**Figure 1:** Soil bulk density (n=5) of alley (yellow) and bush (blue) regimes with increasing years of establishment. Error bars represent $\pm$ 1SD. For a given regime (alley or bush) dissimilar lower-case letters indicate significant ($p \leq 0.05$) differences across the timeseries. At a given timepoint, * indicates a significant difference ($p < 0.05$) between the alley and bush regimes.

from soil compaction. Furthermore, the overall trend of soil bulk density reduction seen over
the course of the 7-year period (**Fig. 1**) is likely a consequence of both increased aggregate
stability and quantity of stable aggregates (**Section 3.2**) alongside increases in soil carbon
stocks (**Section 3.3**), changes  in which are shown to enhance soil physical properties, i.e.
optimising soil bulk density (Topa et al., 2021, Rieke et al., 2022, Kasper et al., 2009).
**3.2 Soil Fractionation**
Soil aggregates that remain stable and resist disaggregation when exposed to water (*water*
*stable aggregates*) are key determinants of soil structure and stability (Whalen et al., 2003).
Soil aggregates can be classified by their formation conditions; *biogenic* (decomposition of
organic matter and action of soil fauna), *physicogenic* (soil physical and chemical processes)



and *intermediate* (a combination of biogenic and physicogenic factors)(Ferreira et al., 2020).
Additionally, land management practice can further influence the formation and stability of
soil aggregates and can significantly alter their formation and destruction   (Lal, 1997, Mikha
et al., 2021).

Stable soil aggregates act as an important indicator of overall soil quality due to their

influence on wider soil properties (Lehmann et al., 2020, Rieke et al., 2022). Aggregates exert
influence over soil bulk density and hydrology, due to the arrangement and make up of soil
structures and pore space (Rieke et al., 2022, Kasper et al., 2009) and can act as a physical
protection for organic matter and carbon (Smith, 2008, Brodowski et al., 2006, Abiven et al.,

2009).

Proportions of WSA and NWSA were seen to change significantly ($p \leq 0.05$) in both the alley

and bush soils (**Fig. 2**). While the sand fraction also observed significant changes ($p \leq 0.05$)
between some of the alley and bush soils (**Fig. 2**), the overall change in sand fraction has been
discounted from further discussion as this fraction cannot be created or altered relative to
the NWSA or WSA fractions.

Soil WSA and NWSA fractions in both the alley soils and bush soils observed opposing trends

with age of establishment. With NWSA in both the regimes reducing in fractional share
significantly ($p \leq 0.05$) over the 7 years of establishment, while the WSA fractional proportion
increased significantly over time ($p \leq 0.05$) (**Fig. 2; Table SI 1** in the supplement). Such changes
were likely due to the effects of halting of soil tillage (*with a decrease in NWSA, and*
*commensurate increase in WSA in the first year of no-till adoption*) and increasing time since
soil disturbance. Furthermore, these shifts in NWSA vs WSA proportions were noted to be
commensurate with soil carbon increases (**Section 3.3**) and SBD decreases (**Section 3.1**),



Collectively these changes may enhance soil aggregate stability and cohesion (Abiven et al.,
2009, Six et al., 2004, Kasper et al., 2009).
NWSA fractions in the alley soils decreased successively with time, from a total of 27.6% in
the control soil to 12.6% in the year 7 soil, with significant reductions (p ≤ 0.05) measured
between the control soil and all regeneratively managed soils (**Fig. 2; Table SI 1** in the
supplement). Additionally, NWSA in the year 7 soil was measured to be significantly lower (p
≤ 0.05) than all other regeneratively managed soils (**Fig. 2; Table SI 1** in the supplement).
In the bush soil, NWSA fractions were also observed to decrease significantly (p ≤ 0.05) in
all regeneratively managed soils relative to the control, ranging between 27.6% in the control
to 15.2% in the year 1 soil (**Fig. 2; Table SI 1** in the supplement). However, this decrease was
not successive, as the greatest reduction was measured in the year 1 soil and increased (not
significantly (p ≥ 0.05)) to then broadly plateau in subsequent years (**Fig. 2; Table SI 1** in the
supplement). Furthermore, no significant differences (p ≥ 0.05) were observed between any
of the regeneratively managed soils.
When compared pairwise significant differences (p ≤ 0.01) between the alley and bush soils
were observed in the year 5 and year 7 soils (**Fig. 2; Table SI 1** in the supplement). NWSA
content of the alley soils was measured to be significantly (P ≤ 0.01) lower than that of the
bushes (15.9% vs. 18.8% in year 5; 12.6% vs. 16.1% in year 7, in the alley and bush soils
respectively) (**Fig. 2; Table SI 1** in the supplement).
Conversely WSA fractions in the alley soils increased broadly with age of establishment,
from 5.8% in the control soil to 16.0% in the year 7 soil, with significant increases (p ≤ 0.05)
measured between the control soil (5.8%) and both the year 5 and year 7 soils (10.3% and
16.0% respectively), (**Fig. 2; Table SI 1** in the supplement). Additionally, the WSA fraction in



year 7 was observed to be significantly greater (p < 0.05) than in all other regeneratively
managed soils (**Fig. 2; Table SI 1** in the supplement).
In the bush soils, the WSA fraction was also observed to generally increase with time, from
5.8% in the control soil to 14.4% in the year 7 soil; with significant increases (p ≤ 0.05)
measured in the year 5 and year 7 soils (11.0% and 14.4% respectively) (**Fig. 2; Table SI 1** in
the supplement). Within the regeneratively managed soils, significant differences (p ≤ 0.05)
were also observed between the year 5 soil and the year 3 soil, and between the year 7 soil
and years 1 and 2 soils (**Fig. 2; Table SI 1** in the supplement). When compared pairwise no
significant differences (p ≥ 0.05) were observed for the WSA content of the alley and bush
soils in each year of regenerative management (**Fig. 2; Table SI 1** in the supplement).

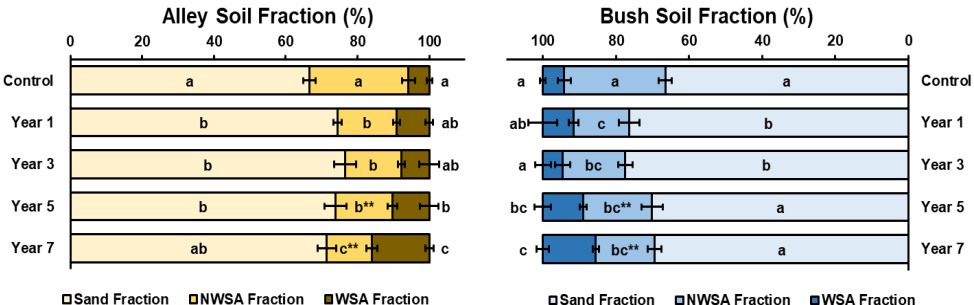

**Figure 2:** *Sand, NWSA, WSA fractions (% total mass)) (n=5) of alley (left) and bush (right) regimes with increasing years of establishment. Error bars represent ± 1SD. For a given regime (alley or bush) dissimilar lower-case letters indicate significant (p ≤ 0.05) differences across the timeseries. At a given timepoint, the \* indicates a significant difference (p ≤ 0.05) between the alley and bush regimes. \*\* indicates a significant difference (p ≤ 0.01), between the alley and bush regimes.*


**3.3 Soil Carbon and Thermal Stability**
Soil organic carbon (SOC) underpins a wide range of ecosystem processes and functions
(Power, 2010, de Groot et al., 2002, Adhikari and Hartemink, 2016, Baveye et al., 2016,
Dominati et al., 2010). The relative stability of the carbon is an underlying feature of the
environmental value and utility of carbon. Indeed, biological function and soil biodiversity rely





heavily upon easily degradable carbon pools with short residence times, while services such
as carbon sequestration and long-term storage rely upon the more stable recalcitrant carbon
pools that can resist degradation (Dell'Abate et al., 2003, De Graaff et al., 2010, Kleber, 2010,
Keenor et al., 2021, Martin and Sprunger, 2022).
SOC was observed to increase in both the alley and bush soils over time (**Fig. SI 2** in the
supplement), with significant increases ($p \leq 0.05$) in the year 5 bush soil (22.3 g kg$^{-1}$ C) and
both the alley and bush soils of year 7 (29.9 g kg$^{-1}$ C and 23.8 g kg$^{-1}$ C respectively) relative to
the control soil (16.6 g kg$^{-1}$ C) (**Fig. SI 2** in the supplement). While increases in SOC were more
pronounced in the alley soils than in the bush soils no significant ($p \geq 0.05$) differences were
observed when compared pairwise (**Fig. SI 2** in the supplement)
Thermal techniques such as thermogravimetric analysis can provide effective means of
characterising organic matter pools in the soil, defining the profile of SOC stability (Plante et
al., 2005, Dell'Abate et al., 2000, Dell'Abate et al., 2003, Plante et al., 2011, Mao et al., 2022).
Furthermore, thermal stability can provide a proxy for biogenic decay and degradation of soil
organic matter and carbon stocks (Plante et al., 2005, Nie et al., 2018, Gregorich et al., 2015,
Plante et al., 2011, Mao et al., 2022).
Total labile and recalcitrant carbon pools were observed to increase in a broadly stepwise
manner over the 7-year period, with marginally more labile carbon than recalcitrant carbon
measured in both alley soils and bush soils and across all years (**Fig. 3**). Additionally, the
content of labile carbon increased significantly ($p \leq 0.05$) in both the alley and bush soils with
time, while no significant differences ($p \geq 0.05$) between recalcitrant carbon pools of either
the alley or bush soils were observed (**Fig. 3**).
Labile soil carbon measured in the alley soils increased broadly stepwise with age of
establishment, with labile carbon increasing in all regenerative managed soils relative to the





control soil (**Fig. 3**). These increases were significant ($p \leq 0.05$) in both the year 5 and year 7
soils relative to the control (increasing from 7.9 g kg$^{-1}$ C $_{labile}$ (control) to 13.6 g kg$^{-1}$ C $_{labile}$,
17.6 g kg $^{-1}$ C $_{labile}$ respectively), i.e., an increase of 9.7 g kg$^{-1}$ C $_{labile}$ (**Fig. 3**). Additionally, the
labile carbon pool measured in the year 7 soil was observed to be significantly greater ($p \leq$
0.05) than that of the year 1 and 3 soils (**Fig. 3**).

In the bush soils, the labile soil carbon pool followed the same trend of broadly stepwise

increase in all regeneratively managed soils relative to the control. Furthermore, significantly
greater ($p \leq 0.05$) carbon stocks were measured in the year 5 and year 7 soils relative to the
control (increasing from 7.9 g kg$^{-1}$C $_{labile}$ to 12.4 g kg$^{-1}$C $_{labile}$ and 13.9 g kg$^{-1}$ C $_{labile}$, respectively)
i.e., an increase of 4.0 g kg$^{-1}$ C $_{labile}$ (**Fig. 3**). Furthermore, significant differences ($p \leq 0.05$) were
measured between regeneratively managed soils (year 5 and 7 vs. year 3; and year 7 vs. year
1) (**Fig. 3**).

When compared pairwise, labile carbon in the alley soil increased by a total of

9.7 g kg$^{-1}$ C $_{labile}$, vs. Increase of 4.0 g kg$^{-1}$ C $_{labile}$ in the bush soil after 7 years of regenerative
management, suggesting enhanced labile carbon stock growth in the alley soils relative to the
bush soils. However, no significant differences ($p > 0.05$) were observed in any given year)
(**Fig. 3**).

Recalcitrant carbon measured in the alley soils increased broadly stepwise with increasing

age of establishment, with all regeneratively managed soils increasing relative to the
conventional control, however none of these increases were significant ($p \geq 0.05$) (**Fig. 3**).

Over the 7 year period recalcitrant carbon in the alley soils increased (not significantly ($p \geq$

0.05)) by 3.6 g kg$^{-1}$ C $_{recalcitrant}$ (from 8.7g kg$^{-1}$ C $_{recalcitrant}$ (control) to 12.3 g kg$^{-1}$ C $_{recalcitrant}$ (year
7 soils) (**Fig. 3**).



In the bush soils, recalcitrant carbon was also observed to generally increase with time (not
significantly ($p \geq 0.05$)). However these increases were smaller than those observed within
the alley soils (**Fig. 3**). Recalcitrant carbon in the bush soil increased (not significantly ($p \geq$
0.05) from 8.7 g kg$^{-1}$ C $_{recalcitrant}$ (control) to 9.9 g kg$^{-1}$ C $_{recalcitrant}$ (year 7) i.e., a difference of
1.2 g kg$^{-1}$ C $_{recalcitrant}$ (**Fig. 3**).
When compared pairwise for labile and recalcitrant carbon stocks in the alley soils and bush
soils, no significant differences ($p \geq 0.05$) were observed between any of the given years.
However, it was observed that both alley and bush soils followed the same trend, with a
greater proportion of both labile and recalcitrant carbon stored within the alley soils (**Fig. 3**).
By year 7, the alley soil was observed to contain a total carbon content of 29.9 g kg$^{-1}$ C (split
as 17.6 g kg$^{-1}$ C $_{labile}$ and 12.3 g kg$^{-1}$ C $_{recalcitrant}$), while the bush soil contained a total carbon
content of 23.8 g kg$^{-1}$ C (split as 13.9 g kg$^{-1}$ C $_{labile}$ and 9.9 g kg$^{-1}$ C $_{recalcitrant}$). In contrast, total
carbon content in the control soil was 16.6 g kg$^{-1}$ C (split as 7.9 g kg$^{-1}$ C $_{labile}$ and
8.7 g kg$^{-1}$ C $_{recalcitrant}$) (**Fig. 3**).

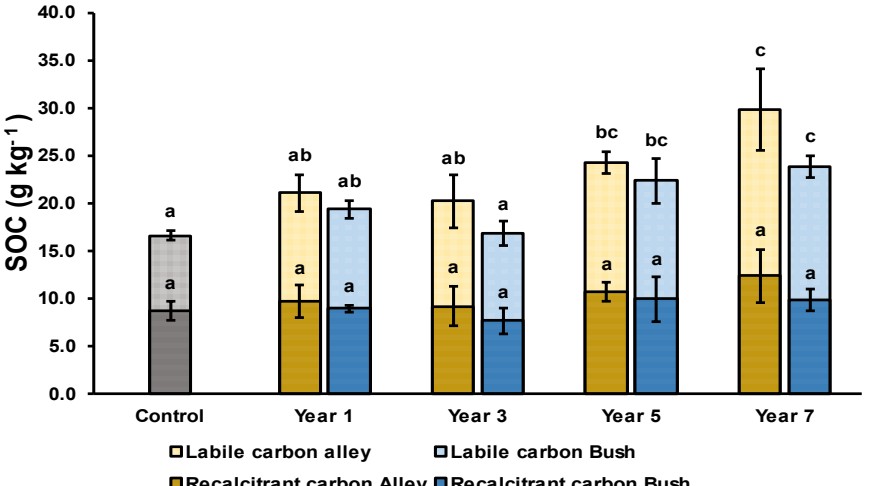

**Figure 3:** SOC split by recalcitrant (hashed) and labile (plain) carbon pools (n=5) in the alleyway (yellow) and bush (blue) regimes. Error bars represent ± 1SD. For a given regime (alley or bush) dissimilar lower-case letters indicate significant ($p \leq 0.05$) differences across the timeseries. At a given
timepoint, * indicates a significant difference ($p < 0.05$) between the alley and bush regimes.



**3.4 Carbon Thermal Stability in Aggregate Fractions**

Total labile and recalcitrant carbon pools, when split by soil fraction, were found to diverge over the 7-year period, with greater proportions of carbon (*both labile and recalcitrant*) observed in the WSA fraction while diminishing in the NWSA fraction with time (**Fig. 4**). It is highlighted that despite their smaller fractional share (**Section 3.2**), WSA were substantially enriched in carbon relative to the NWSA fraction.

Labile carbon in the alley soils was observed to shift between dominance in the NWSA fraction to dominance of the WSA fraction with time, with significant decrease ($p \leq 0.05$) in the NWSA fraction and a non-significant increase ($p \geq 0.05$) in the WSA fraction (**Fig. 4A**).

When analysed by aggregate fraction, the labile carbon pool in the NWSA fraction was observed to significantly decrease ($p \leq 0.05$) with increased time under regenerative management, from 33.7% (control) to 17.5% (year 7). However, no significant differences ($p \geq 0.05$) were measured between the control and the other regeneratively managed soils (**Fig. 4A**).

Within the WSA fraction the labile carbon pool was observed to increase (not significantly ($p \geq 0.05$)) from 45.5% in the conventional control to 61.3% in the year 7 soil (**Fig. 4A**). Initial reductions in the labile carbon pool were observed in year 1 and year 3 relative to the control (reducing to 38.1% in the year 3 soil), before rebounding in years 5 and 7. However no significant differences ($p \geq 0.05$) were observed between any of the soils (**Fig. 4A**).

Labile carbon in the bush soils was similarly observed to shift from dominance in the NWSA fraction to dominance in the WSA fraction with time under regenerative management, culminating in reduced NWSA and increased WSA fraction associated labile carbon by year 7. However, this trend was less pronounced within the alley soil, and no significant differences ($p \geq 0.05$) were observed overall (**Fig. 4B**).



Within the NWSA fraction no significant differences ($p \geq 0.05$) were observed between the
control and any regeneratively managed soil (**Fig. 4B**). Labile carbon initially decreased in year
1 relative to the control (from 33.7% to 24.8%) before converging with the control in years 3
and 5 (33.6% and 33.8% respectively) and subsequently reducing again in year 7 (23.7%) (**Fig.**
**4B**).
In the WSA fraction the labile carbon pool increased (not significantly ($p \geq 0.05$)) between
the control and year 7 soil (45.5% to 54.8%). However these changes were not as substantial
as those observed in the alley soils (**Fig. 4B**). WSA associated labile carbon decreased in the
year 3 soil to 28.2%, while this decrease was not significant ($p < 0.05$) relative to the control,
labile carbon content was observed to rebound significantly ($p \leq 0.05$) from year 3 to year 7
(**Fig. 4B**).
When compared pairwise, a significant difference ($p \leq 0.05$) was observed between the
NWSA fraction of year 5 soil, with 23.7 % of the labile carbon pool contained within the NWSA
fraction of the alley soil relative to 33.8 % in the bush soil; no further significant differences
($p \geq 0.05$) were observed (**Fig. 4 A/B**).
Recalcitrant carbon in the alley soils was also observed to enrich in WSA relative to the
NWSA fractions over time, with the decrease in NWSA being significant ($p \leq 0.05$), while the
increase in WSA was not significant ($p \geq 0.05$) over the 7-year period (**Fig. 4C**).
When analysed by fraction, the recalcitrant carbon pool in the NWSA fraction was observed
to decrease broadly stepwise, with a significant decrease ($p \leq 0.05$) measured between the 7-
year and control soils (from 33.2% to 18.9%) (**Fig. 4C**). Significant differences ($p \leq 0.05$) were
also observed between the year 3 and year 7 soils, where NWSA fraction proportion increased
to converge with the control in the year 3 soil (32.2 %), thereafter decreasing in year 5 and
year 7 (**Fig. 4C**).



In the WSA fraction the recalcitrant carbon pool was observed to increase (not significantly
($p \geq 0.05$)) with time, increasing from 50.1% in the control to 64.5% in the year 7 soil (**Fig. 4C**).
Initial decreases in recalcitrant carbon were observed in the year 1 soil relative to the control
(decreasing (not significantly ($p \geq 0.05$)) to 41.0 %). Thereafter subsequent stepwise increases
in all other regeneratively managed soils were observed (**Fig. 4C**).
Recalcitrant carbon in the bush soils was also observed to increase in the WSA fraction (not
significantly ($p \geq 0.05$)) and decrease (not significantly ($p \geq 0.05$)) within the NWSA fraction
from the control soil to the year 7 soil (**Fig. 4D**).
When analysed by fraction, the recalcitrant carbon pool in the NWSA fraction was observed
to decrease overall by year 7 (from 33.2% in the control to 26.2%). However, no significant
differences ($p \geq 0.05$) were measured between any of the regeneratively managed soils and
the control (**Fig. 4D**).
Within the WSA fraction, recalcitrant carbon was observed to increase overall from the
control to year 7, with initial reductions (not significant ($p \geq 0.05$)) measured in year 1 and 3
relative to the control soil, decreasing from 50.1% in the control to 36.4% in the year 3 soil
(**Fig 4D**). WSA was subsequently observed to increase stepwise to a total of 56.4% in year 7
(not significantly different ($p \geq 0.05$) to the control) (**Fig. 4D**).
When compared pairwise significant differences ($p \leq 0.05$) were observed between the in
the recalcitrant carbon pools of the NWSA fraction in both year 5 and year 7 soils, with 23.9%
and 18.9% stored in the alley soils, vs. 34.1% and 26.2% stored in the bush soils respectively
(**Fig. 4 C/D**).






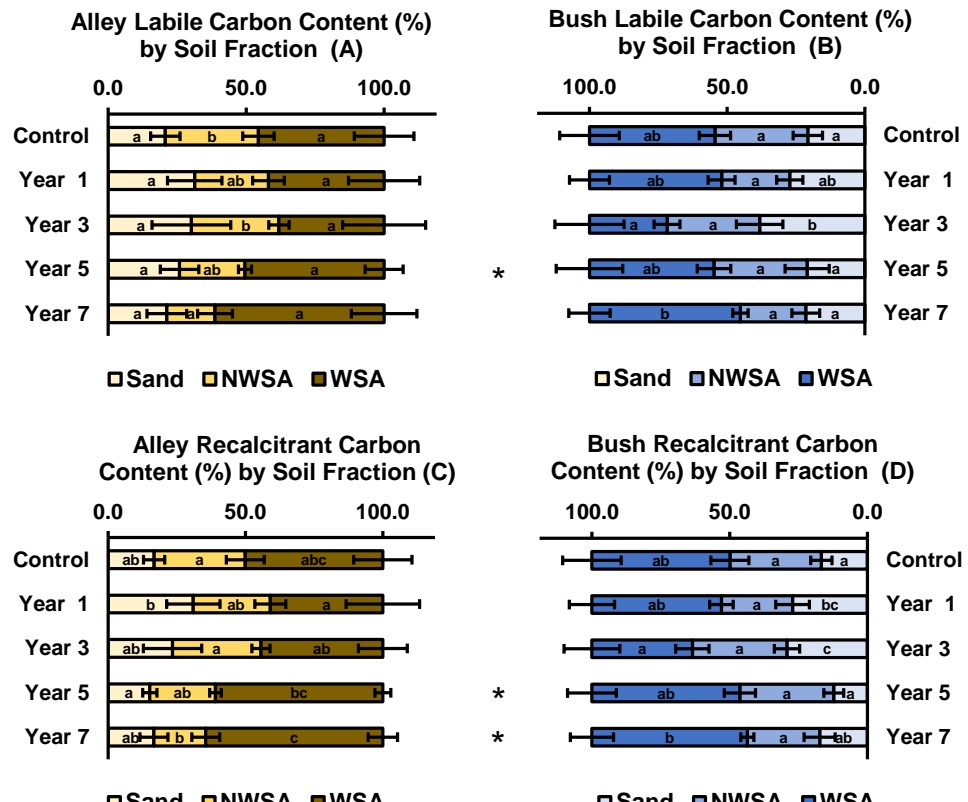

**Figure 4:** Labile (top) and recalcitrant (bottom) SOC split by soil aggregate fraction (Sand, NWSA and WSA) as a total % of soil mass (n=5), of alley (left) and bush (right) soils with increasing years of establishment. Error bars represent $\pm$ 1SD. For a given soil fraction (sand, NWSA, WSA) dissimilar lower-case letters indicate significant (p ≤ 0.05) differences across the timeseries. At a given timepoint, the * indicates a significant difference (p ≤ 0.05) between the alley and bush regimes. ** indicates a significant difference (p ≤ 0.01), between the alley and bush regimes.

**3.5 Aggregate Occlusion of Carbon**


Creation and stabilisation of soil aggregates depend on several key factors, including
climate, soil pH, mineralogy, land management practice, and the
incorporation/decomposition of organic matter content (Wagner et al., 2007, Lal, 1997).
Stable soil aggregates can also confer potentially long-term storage to soil carbon, through
stabilisation and occlusion, physically separating the carbon from its potential vectors of
degradation (Schrumpf et al., 2013, Gärdenäs et al., 2011, Six and Jastrow, 2002, Dungait et



al., 2012, Plante et al., 2011, McLauchlan and Hobbie, 2004, Smith, 2008). As such, stable
aggregate associated labile carbon (occluded carbon) and non-aggregate/NWSA associated
labile carbon (unprotected carbon) can be considered as separate pools where carbon
stability is concerned, despite the inherent lability of both stocks (Six et al., 1998, McLauchlan
and Hobbie, 2004); where decomposition rates of organic matter held within soil aggregates
may be significantly less than non-aggregate associated organic matter, due to the exclusion
of oxygen and soil biota which would otherwise catalyse decomposition (Smith, 2008, Berhe
and Kleber, 2013, De Gryze et al., 2006, Six et al., 1998, Dungait et al., 2012). Additionally,
aggregate size also plays an important role in stabilising carbon, where microaggregates
better protect the soil carbon in the long term (the energy required to break a soil aggregate
being inversely proportional to its size). However, this macroaggregate presence remains
important to both soil structure and the formation mechanics of microaggregates (Six et al.,
2004, McLauchlan and Hobbie, 2004, Dungait et al., 2012, Rabbi et al., 2013). Previous studies
have shown that the carbon contained within soil aggregates may be relatively more labile
than the broader soil environment as a whole, highlighting the efficacy of this physical
protection granted by occlusion within soil aggregates (Six et al., 1998, Dungait et al., 2012,
McLauchlan and Hobbie, 2004).
Stable aggregate occluded carbon considered the stabilised labile carbon stock held within
the WSA fraction (**Section 3.4**), due to the physical protection offered by these aggregate
structures inhibiting the breakdown and decomposition of the carbon stored within.
Conversely unstabilised carbon considered the labile carbon that was not contained within
the WSA fraction (**Section 3.4**), and thus with greater potential for degradation. Additionally,
recalcitrant carbon (**Section 3.3**), was considered stabilised regardless of the soil aggregate
pool in which it was contained due to the relative stability of this carbon fraction.



Occluded carbon in the alley soils was observed to increase broadly stepwise with time,
measuring increased occluded carbon content in all regeneratively managed soils relative to
the conventional control. However, this increase was only significant ($p \leq 0.05$) in the year 7
soil, (increasing from 3.64 g kg$^{-1}$ C to 10.99 g kg$^{-1}$ C in the control and year 7 soil) (**Fig. 5**). In
the bush soil, occluded carbon was observed to follow a similar trend to that in the alley,
increasing significantly ($p \leq 0.05$) from 3.64 g kg$^{-1}$ C in the control to 7.66 g kg$^{-1}$ in the year 7
soil (**Fig. 5**). However, a decrease (not significant ($p \geq 0.05$)) in the occluded carbon content
of the year 3 soil was measured relative to the control soil, reducing to 2.64 g kg$^{-1}$ C, before
rebounding in years 5 and 7 (**Fig. 5**). When compared pairwise, no significant differences ($p \geq$
0.05) were observed between the occluded carbon contents of either the alley soils or bush
soils, with a greater quantity of occluded carbon stored within the alley soils than the bush
soils in all but year 1 (**Fig. 5**).
Unprotected carbon in the alley soils was observed to increase (not significantly ($p \geq 0.05$))
in all of the regeneratively managed soils relative to the control soil. However, this increase
remained broadly similar across all regeneratively managed soils, ranging between 6.4 g kg$^{-1}$
C and 6.7 g kg$^{-1}$ C, compared with 4.2 g kg$^{-1}$ in the control soil (**Fig. 5**). In the bush soil,
unprotected carbon was observed to increase broadly stepwise, with significant increases ($p$
$\leq 0.05$) in the year 3, 5 and 7 soils relative to the control, and increasing to a maximum of 6.6
g kg$^{-1}$ (in the year 5 soil) relative to 4.2 g kg$^{-1}$ in the control soil (**Fig. 5**). When compared
pairwise no significant differences ($p \geq 0.05$) were observed between the regeneratively
managed soils, with unprotected carbon measuring similarly in both the alley soils and bush





soils (**Fig. 5**).

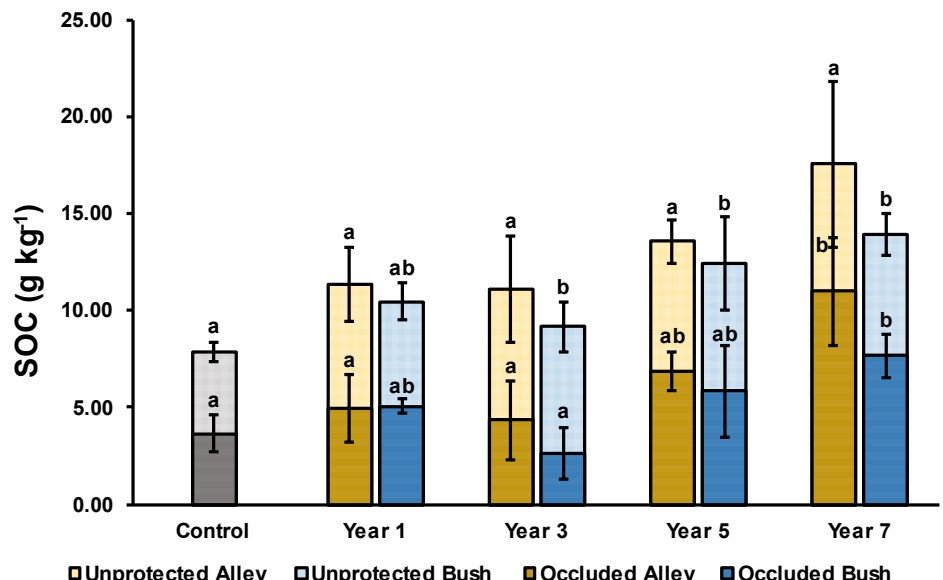

**Figure 5:** Labile SOC split by occluded (hashed) and unprotected (plain) carbon pools (n=5) in the alley (yellow) and bush (blue) regimes. Error bars represent $\pm$ 1SD. For a given regime (alley or bush) dissimilar lower-case letters indicate significant (p ≤ 0.05) differences across the timeseries. At a given timepoint, * indicates a significant difference (p < 0.05) between the alley and bush regimes.


**3.6 Carbon Stability at Field Scale**
Acknowledging proportions of alley and bush soils (60% and 40% of field area, respectively)
and accommodating the influence of SBD (**Section 3.1; Fig. 1**), soil carbon contents (in g C kg$^{-1}$)
$^1$) (**Section 3.3; Fig. SI 2** in the supplement) were converted to carbon stocks (t ha$^{-1}$). These
field scale soil carbon stocks were observed to increase (not significantly (p ≥ 0.05)) by 1.74 t
C ha$^{-1}$ over the 7-year period relative to the control soil (from 21.98 t C ha$^{-1}$ to 23.72 t C ha$^{-1}$)
(**Fig. SI 3** in the supplement).
When considering carbon stocks as split by labile and recalcitrant carbon pools, both were
initially observed to decrease between the control and year 3 soil (**Fig. 6A**). The majority of
this decrease occurred in the recalcitrant carbon stock, decreasing significantly (p ≤ 0.05) from
11.54 t C ha$^{-1}$ to 7.62 t C ha$^{-1}$, while labile carbon stock was observed to decrease gradually



(not significantly ($p \geq 0.05$) from 10.44 t C ha$^{-1}$ to 9.22 t C ha$^{-1}$ (**Fig. 6A**). Following this initial
decrease in both labile and recalcitrant carbon stocks, subsequent yearly increases were
observed in both years 5 and 7, by which point labile carbon stocks were observed to exceed
those in the control (**Fig. 6A**).
Over the full 7-year period recalcitrant carbon stock was observed to decrease (not
significantly ($p \geq 0.05$) to 9.85 t C ha$^{-1}$ (from 11.54 t C ha$^{-1}$), while labile carbon stocks were
observed to increase significantly ($p \leq 0.05$) to 13.87 t C ha$^{-1}$ (from 10.44 t C ha$^{-1}$). Highlighting
that the overall 1.75 t C ha$^{-1}$ increase observed in soil carbon stock over the 7-year period was
comprised entirely of labile carbon (**Fig. 6A ; Fig. SI 3** in the supplement). While recalcitrant
carbon stocks were observed to increase in later years, this rate of increase was less than that
of the labile carbon pool (**Fig. 6A**). However, it is likely that recalcitrant carbon stocks would
recover to the level of the control and possibly increase further with additional time under
regenerative management.  Furthermore, It is likely that the initial decreases observed in both
labile and recalcitrant carbon pools related to soil disturbance and changing inputs when
transitioning from an arable to blackcurrant crop, alongside a soil priming effect from the
increase in labile carbon content increasing the diversity and abundance of soil microbial
communities that promote decomposition (De Graaff et al., 2010, Amin et al., 2021,
Yazdanpanah et al., 2016, Lal et al., 2018). Additionally, it has been observed that significantly
increasing labile carbon inputs to the soil can undermine the stability of recalcitrant carbon
due to this enhanced priming effect (De Graaff et al., 2010), potentially causing the
recalcitrant carbon loss initially observed.
Occluded carbon stocks were observed to increase mildly (not significant ($p \geq 0.05$))
between the control and year 1 soil (from 4.81 t C ha$^{-1}$ to 4.98 t C ha$^{-1}$) , before decreasing
relative to both in the year 3 soil (not significantly ($p \geq 0.05$)) (to 3.23 t C ha$^{-1}$) (Fig. 6B).



Subsequently, occluded carbon stocks were observed to increase in the years 5 and 7 soils (to
5.82 t C ha$^{-1}$ (not significantly ($p \geq 0.05$)), and 8.21 t C ha$^{-1}$ (significantly ($p \leq 0.05$))
respectively). An overall significant ($p \leq 0.05$) increase in the occluded carbon pool between
the control and year 7 soils, almost doubling from 4.81 t C ha$^{-1}$ to 8.21 t C ha$^{-1}$ (Fig. 6B). While
unstabilised carbon was observed to remain broadly consistent across all soils with no
significant differences ($p \geq 0.05$) measured (Fig. 6B). Indeed, unstabilised carbon remained
relatively unchanged between the control and year 7 soil (5.63 t C ha$^{-1}$ and 5.67 t C ha$^{-1}$
respectively). However, a small increase was observed in the year 1 soil following cultivation,
increasing to 6.02 t C ha$^{-1}$, before converging (Fig. 6B). It is highlighted that the significant ($p$
$\leq 0.05$) increase in occluded carbon corresponds to the almost identical increase in labile
carbon measured in the same time period (3.40 t C ha$^{-1}$ and 3.42 t C ha$^{-1}$ respectively) (Fig.
6A/B). As such, it can be concluded that virtually all of the uplift in labile carbon measured
over the 7 year period had been physically protected within the stable aggregate fraction.
This result is important as it confirms regenerative practices have been effective in cultivating
aggregate stability capable of physically protecting what would otherwise be potentially
degradable, labile, carbon. Thus, when viewed as total stabilised carbon (inclusive of
recalcitrant carbon and occluded carbon) a total 1.7 t C ha$^{-1}$ increase (not significant ($p \geq 0.05$)
of potentially sequesterable carbon observed after 7 years of regenerative management
relative to the control (Fig. 6 C).





**Figure 6:** Carbon stock (n = 5) split by recalcitrant carbon (hashed) and labile carbon (plain)(A) and occluded carbon (hashed) and unstabilised carbon (plain)(B); and total stabilised carbon (Green) and unstabilised carbon (plain). Total stabilised carbon considered both recalcitrant and occluded carbon stocks. Error bars represent ± 1SD. Dissimilar lower-case letters indicate significant (p ≤ 0.05) differences across the timeseries.




**3.6 Carbon sequestration**

Efforts to increase soil carbon stocks, through methods such as regenerative agriculture, have become increasingly important strategies to support climate change mitigation (Lal et al., 2004, Smith, 2008, Smith et al., 2020, Soussana et al., 2019, Baveye et al., 2020, Keenor et al., 2021, Lal, 1997, Lal, 2004). However, it is important that we acknowledge not all carbon is equal in terms of its long-term sequestration potential. The results presented herein highlight the important nuances of both recalcitrant carbon pools and the physical protection of carbon (labile and/or recalcitrant) within soil aggregates. Given the physical protection conferred by stable soil aggregates even relatively labile carbon structures may be stabilised and physically protected in the long term as a result of their occlusion from degradative forces; with the aggregate stability governing the carbon residence time rather than its inherent stability (Schrumpf et al., 2013, Gärdenäs et al., 2011, Dungait et al., 2012, Six and Jastrow, 2002, Plante et al., 2011, McLauchlan and Hobbie, 2004)(**Section 3.4; Section 3.5**). While the average mean residence time (MRT) of aggregate stabilised carbon can range from decades to centuries, similarly to that of recalcitrant carbon, the permanence of this carbon can vary greatly between different land use types (as a result of soil management practice) (Six and Jastrow, 2002, Rabbi et al., 2013). As such It is highlighted that carbon protection is only conferred for as long as the carbon is occluded – i.e. activities that damage and destroy soil aggregates (*soil disturbance and ploughing*) can reverse these physical protections and allow for the entry of this carbon to the degradative labile carbon pool from which it had previously been isolated (Pandey et al., 2014, Six et al., 1998, McLauchlan and Hobbie, 2004). Within a no till rotational system, carbon storage within stable aggregates has been observed to range between 27 – 137 years (Six and Jastrow, 2002). Thus providing significant means of stabilising and sequestering carbon in the medium- to long-term, within regeneratively managed



systems (Lal, 1997, Abiven et al., 2009), and potentially on par with that of recalcitrant carbon
stocks (Mao et al., 2022).
For accurate carbon sequestration accounting to be realised, focus must be placed on the
role soil bulk density plays in carbon sequestration calculations; as changes in soil carbon
content often culminate in commensurate changes to the bulk density of a soil (Ruehlmann
and Körschens, 2009, Smith et al., 2020). Simply, as soil bulk density changes, the total volume
that the soil occupies also changes (the total amount of soil remains the same, but its
structure and arrangement in 3D space does not). Where soil bulk density decreases, the mass
of soil per unit volume decreases. Consequently, to increase field-scale carbon stocks
(assessed to a prescribed depth), SOC ($g\ kg^{-1}$) must increase at a greater rate than bulk density
decreases.
In this research, soil bulk density (**Section 3.1**), was observed to decrease with length of
time under regenerative practices, meanwhile soil carbon content (**Section 3.2**) was observed
to increase with time. However, when changes in carbon stocks were considered on a $t\ C\ ha^{-1}$
$^{1}$ basis (with a prescribed soil depth of 7.5cm), carbon stocks did not increase incrementally
with increasing time (**Section 3.6; Fig. SI 3** in the supplement). In effect there was a trade-off,
as the rate of SBD decrease outpaced that of SOC increase. Consequentially, where soil carbon
stocks are considered, while carbon content of the soil increased by ~65% between over the
7 year period (increasing from $16.6\ g\ kg^{-1}$ in the control to $27.5\ g\ kg^{-1}$ after 7 years (alley and
bush soil collectively)), the total field scale increase in carbon stock was only ~8% (increasing
from $21.98\ t\ ha^{-1}$ to $23.72\ t\ ha^{-1}$)(**Fig. SI 3** in the supplement).
Our results highlight the antagonism that exist between SBD and SOC where a prescribed
soil depth is applied to soil carbon stock calculations. Thus, it is arguably more appropriate to
acknowledge the depth of horizon transitions within a soil profile, and where SBD is increasing



(e.g. with time under regenerative practices) to in effect increase the volume of the original
soil, this new soil depth of the horizon should be used in carbon stock calculation.
Yet it is often the case that soil analysis reports provided to farmers do not appreciate these
changes in SBD; rather they present absolute soil carbon content (*%*). As a consequence, the
credibility of both on-farm emissions reductions and creation of soil carbon credits is
undermined, creating low integrity carbon sequestration and may lead to the abandonment
of potentially significant transitional technologies due to a lack of trust. As such, the
standardisation of accountancy methods, (alongside robust validation and verification) is
imperative to restoring confidence and boosting the integrity of soil based carbon
sequestration (Keenor et al., 2021).
Thus, accounting for recalcitrant carbon and total stabilised carbon with respect to SBD,
potentially sequesterable soil carbon was measured to increase over the 7-year period by 1.7
t C ha$^{-1}$ (**Section 3.6; Fig. 6 C**); offering significant benefit and potential to long term carbon
storage at the farm and landscape scale. When calculated against the scale of regenerative
blackcurrant production at Gorgate Farm (50.3 hectares) a total potential of 314 t $CO_2$e could
be sequestered with carbon residence on a decadal timescale.
As perennial plants, soft fruit and orchard crops offer significant opportunities for
investment, engagement, and adoption of regenerative agriculture principles for soil
enhancement and climate change mitigation, due to their low maintenance - long-term
growing habits and the minimal need for soil disturbance. Were the same regenerative
methods as practiced at Gorgate Farm to be applied to all UK soft fruit production (total of
10,819 hectares (DEFRA, 2023)), this could provide a total UK wide sequestration potential of
67,500 t $CO_2$e after 7 years of continuous management, with the potential for further
increases over a longer time period. Whilst this total sequestration after 7 years offers only a



small improvement at a nationwide scale, this could be achieved with minimal changes to
current soft fruit production management practice. Furthermore,
**4. Conclusion**
The results of this research highlight the potential for regenerative agriculture practices to
increase SOC, increase the proportions of WSA, enrichment and physically protect labile
carbon within these aggregates and thus afford opportunity for long-term carbon
sequestration as stabilised carbon stocks. However, our results also bring to the fore
important factors relating to soil carbon stock assessment. In particular, the antagonism
between SBD decreasing at a rate greater than SOC increases; this creating a trade-off where
soil carbon stocks are calculated to a standard prescribed depth. Further research and
practical guidance is needed to enable more robust soil carbon stock assessment that
acknowledges i) a full pedogenic soil horizon, ii) the inherent reactance of SOC, and iii) the
proportion of SOC physically protected by association with soil aggregates.
**Authorship contribution**
Reid was the Principal Investigator and Keenor the Senior Researcher for this research. Together
Keenor, Reid and Lee undertook the investigation planning and fieldwork. Laboratory work was led by
Keenor with assistance in preliminary laboratory study and WSA method development from Lee.
Keenor undertook the soil data and carbon stability analysis, statistical analysis, literature review, and
the drafting of the manuscript. Keenor and Reid undertook review and editing to deliver the final
manuscript.
**Acknowledgments**
This Research was supported by the Natural Environment Research Council and ARIES DTP [grant
number NE/S007334/1] with additional support provided by Greenworld Sales Ltd, Norfolk.
**Competing Interests**
The authors have no competing interests to declare.



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
