# Peer review of "Physical Protection of Soil Carbon Stocks Under Regenerative"

_EGUsphere, 2024_

## Author Response (AR1)

Dear Katerina,

**egusphere-2024-4029**

**Physical Protection of Soil Carbon Stocks Under Regenerative Agriculture**

Please find attached a revised copy of the manuscript. In the tables below we account a summary of our major changes to the manuscript in line with your and the reviewers' comments, and a more detailed table of the reviewers' comments and subsequent actions we have taken.

In addition, both a tracked and clean version of the revised manuscript are attached.

Table 1 - Summary of the Headline Changes

| Comments and Suggestions                  | Actions taken                                            |
|-------------------------------------------|----------------------------------------------------------|
| Provide further clarity regarding         | Further clarity given in the methods section             |
| experimental details.                     | regarding the arable control field management,           |
|                                           | including cultivation depth. Details of the fallow and   |
|                                           | cover crops mix has been given in the supporting         |
|                                           | information with a link to table SI 1 in text. Dates for |
|                                           | soil sampling have been clarified in the methods.        |
|                                           | Further information regarding fertiliser type, use and   |
|                                           | quantities has been given in the text for both the       |
|                                           | arable control and the blackcurrant fields.              |
| Figure 1 from the supplement should       | This figure has been moved from the supplementary        |
| be included within the main text for      | material to Figure 1 of the main manuscript.             |
| clarity.                                  |                                                          |
| If required, an additional table within   | An additional table as a glossary of terms has been      |
| the supplementary section can be          | added to the supplementary section (Table SI2)           |
| included which details each type of       | defining key terms and their origins                     |
| carbon discussed in the investigation     |                                                          |
| and their origins. A glossary of terms if |                                                          |
| you will.                                 |                                                          |
| Better balance the content between        | Text has been reviewed to remove unnecessary             |
| the introduction and discussion           | literature from the discussion section into a relevant   |
| sections, being mindful of content        | part of the introduction to improve context,             |
| that may be repetitive or                 | irrelevant literature and discussion– or that which is   |
| inappropriately placed within the         | repetitive has been removed. Furthermore, where          |
| discussion section. With further in-      | necessary to improve clarity further context has         |
| depth thought given to the content of     | been given, or ambiguous phrasing removed.               |
| the manuscript in accordance with         |                                                          |
| the suggestion of the editors and         |                                                          |
| reviewers.                                |                                                          |
| Further clarity requested as to the       | This section of text within the abstract has now been    |
| statements of 'insight towards the        | edited to remove reference to the mechanisms of          |
| mechanisms of soil carbon                 | soil carbon stabilisation, removing this ambiguity,      |
| stabilisation', given the lack of data    | with a renewed focus placed upon the importance of       |
| regarding additional soil and             | soil physical protection of carbon for potential long-   |
| microbial properties and their            | term carbon storage – as such this is more in line       |
| potential influence.                      | with the concept of the manuscript more broadly.         |

| Review grammar, spelling and | Typos and missing text have been checked and  |
|------------------------------|-----------------------------------------------|
| readability of the text.     | amended where necessary throughout the        |
|                              | manuscript, and units have been standardised. |

We hope that the amendments outlined above, in addition to the changes made to the text of the paper, are satisfactory to correcting the identified limitations of the previous manuscript version. We highlight that the changes better justify the purpose of the study, while removing ambiguities and conjecture in the discussion.

We sincerely thank you for your insights and your time.

Kind regards,

Sam G. Keenor, and Brian J. Reid

**Reviewers Comments and Actions Taken**

**Reviewer 1**

**Comment**

First, the methods were not introduced clearly in the current manuscript, making it difficult to judge the results. For example, what were the management practices (fertilizer, tillage, and so on) that were used for the conventional farming? What were the grazing cover crops that were planted on the alleyways? What were the amount and nutrient content of the sprays of compost tea and organic fertilizer? Which year was the soil sample collected? There were many different soil carbon types in this manuscript (e.g. labile and recalcitrant carbon, occluded carbon, stabilised carbon, WSA, and NWSA); however, the authors failed to separate them clearly. It is hard for me to digest so many different terminologies.

Second, the main text should be improved seriously. For example, many parts in the Results and Discussion section were not about "results and discussion" but were about "background", which should be moved to the introduction section. In addition, more information about the field design and methods should be included in the abstract; otherwise, it is difficult for readers to know the meaning of "alley soil and bush soil".

**Actions Taken**

- Text has been cleaned up to ensure that section headings are appropriately placed above the text and not on adjacent pages.
- More information has been presented detailing the management practices of both the arable control field and the blackcurrant fields
  - This is inclusive of fertiliser types and treatments
  - The types and quantities of cover crops planted
- Further clarity on the dates of soil sampling has been included in the text
- Issues pertaining to the separation of specialist terms and carbon/aggregate types and their definition and use have been further clarified. The text has been combed to ensure that only one specific title be given to each carbon type or aggregate type to improve clarity. In addition, further explicit explanation of the different terms is given in the text as well as a supplementary table (table SI 1) to provide a glossary of terms and better define the specific term used and its origin/purpose in the study.
- Results and discussions section have been reviewed and edited to remove unnecessary background content, or repetitive content. Some of this remaining content has been moved to the introductory section, or within the discussion to re-balance the text and place this in better context, improving the flow and direction of the discussion.
- Additional information regarding the field design and methods has been given in brief in the abstract, improving clarity, and in more depth in the methods section, aligning with the actions taken regarding the reviewer's previous comments.
- Clarity provided in the abstract for "alley soil and bush soil" with greater contextual relevance.

Third, in the abstract section, the authors stated that "This research provides valuable insights into the mechanisms of soil carbon stabilisation under regenerative agriculture practices." However, I am not sure if they really unravel the mechanisms because they did not measure other soil variables (e.g. soil nutrient content and soil microbial parameters), which can be used to explain the results. Moreover, the "Results and Discussion" section was mainly introducing background and describing the results. Instead, more deep discussions and measurements should be included to explain the interesting results (e.g. when compared to control treatment, why did total carbon stock decreased in the short term and then increased to the similar level like the control treatment in the long term). Does this mean regenerative agriculture must be conducted for a long time? Otherwise, croplands would lose soil carbon?

- This sentence has been removed from the abstract, following consideration from the authors in agreement with the reviewer. In its place and edit has been made to better justify the importance of stable aggregates for soil carbon stabilisation, while removing a point of conjecture.
- Results and discussion section has been rebalanced as noted above.
- Further in-depth discussion regarding some of the measurements has been given where relevant, including where the reviewer notes changes in the carbon stock in the short term, and the subsequent impacts of regenerative agriculture.

**Reviewer 2**

**Comment**

The article lacks clarity regarding crucial details such as the experimental set-up (number of fields, number of samples and replicates, etc.) and the presentation of results. Figure 1 from the supplement should be included in the article to provide a clearer picture of the set-up; a map of the experiment could also bring some clarity. It would also be interesting to discuss the prior presence of blackcurrant fields in years 5 and even 3: did the introduction of new plants lead to a return to a state of conventional management, thereby 'resetting' the regenerative agriculture counter to 0 years?

**Actions Taken**

- Further details regarding the methods and experimental set up have been included to improve the clarity of the investigation
  - this has included explicit reference to the number of fields and their age, the number of samples and replicates.
- Figure 1 from the supplement has now been included in the main body of text for additional clarity of the cropping history and experimental set up.
- Discussion of the prior presence of blackcurrants in the year 5 field has been made, along with a detailed explanation as to its valid inclusion within the context of the wider investigation (no soil disturbance), and the importance of the intrinsic soil disturbance during replanting has initiated a 'new cycle' of regenerative management. However, the phrasing of "reset" has been purposefully avoided so as to not be misconstrued.

The results should be discussed further, and the text should be better organized between what is general introduction of the topic, results, and discussion. Even if results and discussion can be addressed altogether, they should be clarified and strengthened.

- Results and discussions section have been reviewed and edited to remove unnecessary background content, or repetitive content. Some of this remaining content has been moved to the introductory section, or within the discussion to re-balance the text and place this in better context, improving the flow and direction of the discussion.

Overall, the article is not easily readable. Check for typos, missing verbs or parts (abrupt end L.631 for example). Indications regarding p-values appear repeatedly throughout the text (as you defined two levels of confidence) without really bringing relevant information; the p-values could appear on your graphs but not in text for instance, with only mentions to 'significant' or 'not significant'.

- The manuscript has been reviewed for errors and typos, missing verbs and erroneous sentences, and corrected where necessary.
- No changes have been made to the presentation of statistical data, while the authors appreciate the viewpoint of the reviewer, our view is for improved clarity that these should remain in place. Additionally, further clarity regarding the two levels of significance used within the study has been provided in the methods section.

**Appendix - Reviewer Comments as received**

**Reviewer 1 Comment:**

This manuscript aims to determine the effects of regenerative agriculture on soil carbon stocks. The authors found that regenerative agriculture did not increase recalcitrant carbon stocks but did significantly increase labile carbon stocks. A lot of work has been done, and some results are interesting. However, I do not think the current version is suitable for publication in SOIL because of several weaknesses. Please see below for my main concerns:

First, the methods were not introduced clearly in the current manuscript, making it difficult to judge the results. For example, what were the management practices (fertilizer, tillage, and so on) that were used for the conventional farming? What were the grazing cover crops that were planted on the alleyways? What were the amount and nutrient content of the sprays of compost tea and organic fertilizer? Which year was the soil sample collected? There were many different soil carbon types in this manuscript (e.g. labile and recalcitrant carbon, occluded carbon, stabilised carbon, WSA, and NWSA); however, the authors failed to separate them clearly. It is hard for me to digest so many different terminologies.

Second, the main text should be improved seriously. For example, many parts in the Results and Discussion section were not about "results and discussion" but were about "background", which should be moved to the introduction section. In addition, more information about the field design and methods should be included in the abstract; otherwise, it is difficult for readers to know the meaning of "alley soil and bush soil".

Third, in the abstract section, the authors stated that "This research provides valuable insights into the mechanisms of soil carbon stabilisation under regenerative agriculture practices." However, I am not sure if they really unravel the mechanisms because they did not measure other soil variables (e.g. soil nutrient content and soil microbial parameters), which can be used to explain the results. Moreover, the "Results and Discussion" section was mainly introducing background and describing the results. Instead, more deep discussions and measurements should be included to explain the interesting results (e.g. when compared to control treatment, why did total carbon stock decreased in the short term and then increased to the similar level like the control treatment in the long term). Does this mean regenerative agriculture must be conducted for a long time? Otherwise, croplands would lose soil carbon?

**Reviewer 2 Comment:**

This paper explores the impact of regenerative management on soil aggregates and carbon pools. The samples originate from blackcurrant fields (bushes and alleys) with duration from 1 to 7 years, and a control plot still under conventional management. Many analyses were conducted (soil bulk density, stable and non-stable aggregate contents, soil carbon contents and stocks, and carbon stability). The main observation is a shift towards more stable aggregates and an increase of the labile carbon pool under regenerative management, with little effect on the recalcitrant pool.

The aim of this research and the results are interesting. Although I think the article is not acceptable at this stage, I would recommend resubmitting the manuscript after significant

changes and thorough proofreading to enhance the work that has been conducted. Some of the elements that should be reconsidered follow.

The article lacks clarity regarding crucial details such as the experimental set-up (number of fields, number of samples and replicates, etc.) and the presentation of results. Figure 1 from the supplement should be included in the article to provide a clearer picture of the set-up; a map of the experiment could also bring some clarity. It would also be interesting to discuss the prior presence of blackcurrant fields in years 5 and even 3: did the introduction of new plants lead to a return to a state of conventional management, thereby 'resetting' the regenerative agriculture counter to 0 years?

The results should be discussed further and the text should be better organized between what is general introduction of the topic, results, and discussion. Even if results and discussion can be addressed altogether, they should be clarified and strengthened.

Overall, the article is not easily readable. Check for typos, missing verbs or parts (abrupt end L.631 for example). Indications regarding p-values appear repeatedly throughout the text (as you defined two levels of confidence) without really bringing relevant information; the p-values could appear on your graphs but not in text for instance, with only mentions to 'significant' or 'not significant'. Also, a genuine thought: could some results be non significant due to the small numbers of samples? Would other tests be better at handling this issue?

---

## Referee Report (RR1)

This paper explores the impact of regenerative management on soil aggregates and carbon pools in blackcurrant fields with different durations ranging from 1 to 7 years. The control is a nearby plot under conventional management. The main observation is an increase of the water stable aggregates and the labile carbon pool under regenerative management. Little effect is observed on the recalcitrant pool, although this may be a result of the 'short' duration of the experiment.

The subject and its implications are important and well detailed. The study site and the methods are described. Results and Discussion are presented together. A lot of work has been conducted here, thus many results are presented, which may be detrimental to the clarity of the paper: while explained, the numerous notions (WSA/NWSA, labile/recalcitrant, aggregates, thermal stability in aggregates, bush/alley soils, several years...), make the article uneasy to read.

I suggest some data appear only in Supplementary Information (e.g. bush soils detailed in the paper and alley soils only in SI). You could also consider making a shorter article, focusing more on certain results, and backing it up by a data paper for the rest of the results.

Here are a few specific remarks:

L24: it would be good to reformulate this sentence, as the 'proportion of NWSA: WSA' is not really 'increasing' (only the % of WSA is indeed increasing).

L71: non verbal sentence, I guess it is rather the end of the previous sentence. Same for L81.

L88: soil properties regulation

L198: 'that which passed through the 63 µm sieve after the HMP treatment'

L222: what is expected between 700 and 1.000°C? (Or – why not stop heating before 1.000°C?)

L252: any idea why the lowest global SBD is in year 3 treatment and not year 7?

L262: while it does not fully change the overall observation, I would not say 'generally decrease with time'

L318: there is no year 2 soil, I guess it is 3.

L342-347: So, recalcitrant carbon content increases with time, but not significantly? This paragraph is not totally clear.

L352: if you mention the increase for year 7, you should also mention the one for year 5 as you talk about both years. Same for L359.

General: check for punctuation, spelling and grammar. Several non-verbal sentences are still found along the text.

---

## Author Response (AR2)

Dear Katerina,

Please find below a list of the intended actions. These reflect the comments of yourself and the most recent reviewers.

Regarding Reviewer 3's comments, we have obliged where we can. However, given the comments received it would appear the reviewer may have not reviewed the corrected version of the manuscript. Their comments do not appear to reflect the corrected version of the manuscript most recently returned to you.

Some of their comments e.g. "The concept of biochemical recalcitrance [...] is introduced in lines 67–68", and ", in section 2.1 I do not see systematic information on the type and amount of fertilization and organic matter inputs [...] it makes the reader wonder on the fertilization applied in the conventional field (no info given), and crop residue management in the regenerative field (no info given) [..]" relate to information that is included in the version of the paper most recently returned to you. We also note that Reviewer 3 has not reviewed the entire paper and may have missed the additional details they request, which are present in the discussion. However, we have actioned their comments where appropriate.

In addition, please also find attached a clean and tracked-changes copy of the manuscript reflecting these changes as outlined below.

Yours sincerely,

Sam Keenor

**Reviewers/Editors Comments and Suggested Edits**

Authors focus on assessing SOC stocks in the field, which is the amount of SOC stored across the soil profile at minimally 30 cm depth per unit area, according to the IPCC guidelines (Penman et al 2003, "Good practice guidance for land use, land-use change and forestry"). Especially in the case of reduced tillage practices as part of the transition towards regenerative farming, assessing SOC stocks at multiple depths is relevant. Reduced tillage is known to 'concentrate' SOC in the upper soil layers, as this SOC is no longer mixed with lower soil layers. Therefore, SOC contents at lower soil layers usually is 'diluted' with reduced tillage, so that SOC stocks across the entire soil profile may not be that different after all. Therefore, SOC stocks should have been measured at lowers depths, preferable even lower than 30 cm as subsoil processes are very different from topsoil processes (Rovira et al, 2022, CATENA). In this study, SOC is only measured at 0-7.5cm depth (line 144, and I think this is insufficient given the information above.

Authors mention multiple times the carbon sequestration potential when interpreting the results (e.g. lines 20, 26, 563). However, it is important to distinguish carbon sequestration (truly capturing CO2 from air into soil) vs. carbon reallocation (moving carbon from one field to another), see Leifeld et al 2013, PNAS. To assess whether the study design can inform the carbon sequestration potential, it is important to know the organic matter inputs to the field. However, in section 2.1 I do not see systematic information on the type and amount of fertilization and organic matter inputs. The use of organic fertilizer and compost tea for regenerative farming is mentioned for regenerative farming, and

While the reviewer identifies that the depth of soil carbon stocks does have an impact upon the total carbon sequestration potential and any 'dilution effects', the limit of 7.5cm depth for measurement was a methodological constraint.

However, this measurement depth was kept constant between both the regeneratively managed fields and the conventional control – thus we highlight that these effects would be equivalent in both cases, and changes in the soil carbon content of each soil would be valid to the depths at which we sampled for comparison.

Given the scope of this project and the inability to re-sample we suggest no action.

We have however, included a statement in the manuscript regarding the point raised by the reviewer as a limitation of the research. This statement is tethered to the citations provided by the reviewer.

Reference to the type and amount of fertiliser applied to both the conventional control fiend and the blackcurrant fields are explicitly mentioned in the text of section 2.1. Furthermore, no additional organic matter has been added to the fields during the course of the experiment.

Additionally, reference to the depth of cultivations (in the control soil) is explicitly stated in section 2.1

In making revisions we have clarified the mechanism of soil carbon stock build. These being 1. Mitigated organic carbon mineralisation due to low soil disturbance and 2. Increased carbon flows into the soil from blackcurrant

stubble re-incorporation for conventional farming. However, it makes the reader wonder on the fertilization applied in the conventional field (no info given), and crop residue management in the regenerative field (no info given). Given the application of organic fertilizer in the regenerative fields, I would say that at least part of the increased SOC contents derive from carbon reallocation rather than sequestration.

Moreover, I also miss more information given the tillage operations (e.g. depth), given its importance for SOC transfer across the soil profile.

residues and the roots of the blackcurrant bushes. In both cases these carbon flows are likely higher than in the arable control, where crop biomass is harvested and exported out of the field.

The concept of biochemical recalcitrance of carbon is outdated, except for pyrogenic carbon (Schmidt et al, 2011, Nature), unless argued otherwise. However, this concept is introduced in lines 67-68 without reference, and interpreting the TGA results as currently done (labile vs. recalcitrant SOC) is only supported by one reference without theoretical explanation. It also makes me wonder whether physically protected SOC within aggregates is part of this recalcitrant fraction? I wonder why it is not chosen to use the t50 of TGA that accounts for the continuum of soil carbon stability (Lehmann et al 2015, Nature)?

I believe deviating from these common concepts and instead use a simplified 2-pool classification of labile vs recalcitrant SOC needs to be much more substantiated, and based on theoretical stabilization mechanisms of SOC, as is done for the distinction of POM-MAOM by size or density fractionation by Lavallee and Cotrufo.

Further content has been included in the more recent iterations of the manuscript to better explain the concepts of carbon recalcitrance.

"I wonder why it is not chosen to use the t50 of TGA that accounts for the continuum of soil carbon stability (Lehmann et al 2015, Nature)?". We are unclear on what the reviewer is driving at here and what they expect by way of an action.

Reference to the Schmidt et al., 2011 paper has been made where relevant in the text.

With regards to POM-MOAM the reviewer makes a valid point. We highlight that the conceptual framework of the paper is aligned to this, in so much as our results explicitly define carbon pools and their recalcitrance in non-stable and stable aggregates. however, given the soil fractionation methodology, there is no clear way to define explicitly what would constitute POM vs. MAOM in the samples. A statement has been added to the methodology section sustained by the citation provided by the reviewer has been included to explain this, although we highlight the methodology used is a

well acknowledged standard. Furthermore, reference has been made in the conclusions regarding opportunities for further research. Section 2.6: Given the changed tillage This is a key point made in the discussion regime that potentially alters bulk section of the text, and was indeed the density, I believe that not the fixed depth purpose of this methodological decision. method but the equivalent soil mass Thus, arguing the relevance of the method should have been used to equivalent soil mass method, highlighting calculate SOC stocks from SOC content that due to changes in soil bulk densities (Von Haden et al 2018, GCB) following changes in soil tillage regimes and sample depth. Suggesting this as a superior method for future work based on the results and design of this experiment. No further action required. Section 2.7: what about meeting the These are standardised and robust assumptions for these models, have they statistical tests. Pre-processing of data been checked, and can hence the was conducted to confirm normality of statistical results be trusted? data prior to application of statistical analysis. No further action required. Based on the observations, I recommend Considering the first round of reviewer to reject this manuscript in its current comments we restructured the text. This form for publication. However, I do reframed the text to acknowledge changes to the SOC stock and quality believe that the experiment and results are super interesting and can be used if and takes the focus away carbon the story is not focused on assessing sequestration alone. SOC stocks and carbon sequestration, but on assessing the changes in SOC In further revision in response to the quality and aggregate stability in top soil second round of reviewers' comments, we will seek to make this direction after a system transformation towards regenerative farming, acknowledging the clearer. However, some focus on continuum of SOC stability and providing sequestration potential will be retained more quantitative management as we believe this is an important aspect information relevant for SOC to include. characteristics. **Next reviewer** A lot of work has been conducted here, We feel that removal of either the alley or thus many results are presented, which bush regime data into the supplementary, may be detrimental to the clarity of the while shortening the text, would serve to paper: while explained, the numerous create confusion. Outcomes for the bush notions (WSA/NWSA, labile/recalcitrant, vs alley regimes are markedly different. To aggregates, thermal stability in pick one over the other for inclusion in

the main text would be misleading.

aggregates, bush/alley soils, several

| January Information (e.g. bush soils detailed in the paper and alley soils only in SI). | Further, where SOC stocks are calculated the proportional area of bush and alley, along with the SOC concentration and soil bulk density, are conflated. Moving either the bush or alley data to the SI will remove a lot of transparency in the reporting.  This said, in further revision to the manuscript we will seek to further reduce unnecessary information and where appropriate truncate text and discussion to improve readability and comprehension of the text. |
|-----------------------------------------------------------------------------------------|-------------------------------------------------------------------------------------------------------------------------------------------------------------------------------------------------------------------------------------------------------------------------------------------------------------------------------------------------------------------------------------------------------------------------------------------------------------------------------|
| L24: it would be good to reformulate this                                               | Sentence will be reformulated to reflect                                                                                                                                                                                                                                                                                                                                                                                                                                      |
| sentence, as the 'proportion of NWSA:                                                   | this comment.                                                                                                                                                                                                                                                                                                                                                                                                                                                                 |
| WSA' is not really 'increasing' (only the %                                             |                                                                                                                                                                                                                                                                                                                                                                                                                                                                               |
| of WSA is indeed increasing).                                                           |                                                                                                                                                                                                                                                                                                                                                                                                                                                                               |
| L71: non verbal sentence, I guess it is                                                 | Text and grammar have been corrected.                                                                                                                                                                                                                                                                                                                                                                                                                                         |
| rather the end of the previous sentence.                                                |                                                                                                                                                                                                                                                                                                                                                                                                                                                                               |
| Same for L81.                                                                           |                                                                                                                                                                                                                                                                                                                                                                                                                                                                               |
| L88: soil properties regulation                                                         | Text and grammar have been corrected.                                                                                                                                                                                                                                                                                                                                                                                                                                         |
| L198: 'that which passed through the 63                                                 | Text and grammar have been corrected.                                                                                                                                                                                                                                                                                                                                                                                                                                         |
| μm sieve after the HMP treatment'                                                       |                                                                                                                                                                                                                                                                                                                                                                                                                                                                               |
| L222: what is expected between 700 and                                                  | This was done in accordance with                                                                                                                                                                                                                                                                                                                                                                                                                                              |
| 1.000°C? (Or – why not stop heating                                                     | following the methods and temperature                                                                                                                                                                                                                                                                                                                                                                                                                                         |
| before 1.000°C?)                                                                        | windows described in Mao et al., (2021)                                                                                                                                                                                                                                                                                                                                                                                                                                       |
|                                                                                         | with only the carbon measured before                                                                                                                                                                                                                                                                                                                                                                                                                                          |
|                                                                                         | 700°c, reports suggest that between                                                                                                                                                                                                                                                                                                                                                                                                                                           |
|                                                                                         | ~700-1000°C predominantly inorganic                                                                                                                                                                                                                                                                                                                                                                                                                                           |
|                                                                                         | carbon sublimes from the sample (Mao et                                                                                                                                                                                                                                                                                                                                                                                                                                       |
|                                                                                         | al., 2021). Inclusion of this previously                                                                                                                                                                                                                                                                                                                                                                                                                                      |
|                                                                                         | unmentioned fraction now mentioned in                                                                                                                                                                                                                                                                                                                                                                                                                                         |
| 1050 11 1 11 1 1 1000                                                                   | the methods section.                                                                                                                                                                                                                                                                                                                                                                                                                                                          |
| L252: any idea why the lowest global SBD                                                | This difference is likely relating more to                                                                                                                                                                                                                                                                                                                                                                                                                                    |
| is in year 3 treatment and not year 7?                                                  | the underlying soil physiology and soil                                                                                                                                                                                                                                                                                                                                                                                                                                       |
|                                                                                         | stoniness than to management practice,                                                                                                                                                                                                                                                                                                                                                                                                                                        |
|                                                                                         | (I.e soil texture and aggregate fraction                                                                                                                                                                                                                                                                                                                                                                                                                                      |
| 1.000 while it does not fully about the                                                 | (Figure SI 3)) now highlighted in text.                                                                                                                                                                                                                                                                                                                                                                                                                                       |
| L262: while it does not fully change the                                                | Text revised.                                                                                                                                                                                                                                                                                                                                                                                                                                                                 |
| overall observation, I would not say                                                    |                                                                                                                                                                                                                                                                                                                                                                                                                                                                               |
| 'generally decrease with time'                                                          | Tout vouised                                                                                                                                                                                                                                                                                                                                                                                                                                                                  |
| L318: there is no year 2 soil, I guess it is 3.                                         | Text revised.                                                                                                                                                                                                                                                                                                                                                                                                                                                                 |
| L342-347: So, recalcitrant carbon content                                               | Text improved.                                                                                                                                                                                                                                                                                                                                                                                                                                                                |
| increases with time, but not significantly?                                             |                                                                                                                                                                                                                                                                                                                                                                                                                                                                               |
| This paragraph is not totally clear.                                                    | Toyt improved reflecting the higher                                                                                                                                                                                                                                                                                                                                                                                                                                           |
| L352: if you mention the increase for year                                              | Text improved reflecting the higher                                                                                                                                                                                                                                                                                                                                                                                                                                           |
| 7, you should also mention the one for                                                  | improvement in the final year.                                                                                                                                                                                                                                                                                                                                                                                                                                                |

| year 5 as you talk about both years. Same                                                                                                                                                                                                                                                                                                                                                                                                                                                                                                                                                                                |                                                                                                                                                                                                                                                                                                                                                                         |
|--------------------------------------------------------------------------------------------------------------------------------------------------------------------------------------------------------------------------------------------------------------------------------------------------------------------------------------------------------------------------------------------------------------------------------------------------------------------------------------------------------------------------------------------------------------------------------------------------------------------------|-------------------------------------------------------------------------------------------------------------------------------------------------------------------------------------------------------------------------------------------------------------------------------------------------------------------------------------------------------------------------|
| for L359.                                                                                                                                                                                                                                                                                                                                                                                                                                                                                                                                                                                                                |                                                                                                                                                                                                                                                                                                                                                                         |
| Please check the numbering of your figures before the next revision since Figure 1 exists twice                                                                                                                                                                                                                                                                                                                                                                                                                                                                                                                          | Figures, Tables and figure legends will be checked throughout to reflect correct numbering position and content in the text.                                                                                                                                                                                                                                            |
| General: check for punctuation, spelling and grammar. Several non-verbal sentences are still found along the text.  move surplus discussion to the supplement to aid readability                                                                                                                                                                                                                                                                                                                                                                                                                                         | Text and grammar will be reviewed throughout to improve readability and comprehension, and correct mistakes.  In further review of the manuscript we have further reduced unnecessary or superfluous information from the introductory literature review and discussion sections of the paper, and improved the conciseness, readability and comprehension of the text. |
| Remove the word "associated" from NWSA and WSA in Table S1, to match how it is used throughout the text. Also, consider adding the word "thermally" to labile and recalcitrant carbon to specify the operational definition. In fact, potential caveats of this operational definition and other mechanisms governing "recalcitrance" (as brought up by Reviewer #3 and reviewed in Schmidt et al. Nature 2011) should be mentioned briefly when the method is first introduced (line 134). This is all nicely done in the title and text of section 3.3, but could be added briefly to the introduction and throughout. | Text in the SI table has been amended to exclude associated and include thermally, additionally, reference to thermal lability/recalcitrance has been noted in the introduction and methods text, with a qualifying note of language going forward.  Additionally, these have been noted in all relevant figures.                                                       |
| Spell out abbreviations in figure captions to make it easier for figures to be understood on their own.                                                                                                                                                                                                                                                                                                                                                                                                                                                                                                                  | Figures, Tables and figure legends will be checked throughout to reflect correct numbering position and content in the text.                                                                                                                                                                                                                                            |
| Line 46 – consider replacing the semi-
colon with a comma or rewording into
two sentences.                                                                                                                                                                                                                                                                                                                                                                                                                                                                                                                         | Change made.                                                                                                                                                                                                                                                                                                                                                            |
| Line 59 - combine this sentence with the paragraph below and remove first comma. Also, consider splitting the following paragraph which is quite long.                                                                                                                                                                                                                                                                                                                                                                                                                                                                   | Change made.                                                                                                                                                                                                                                                                                                                                                            |
| Line 132 – the word respectively is not needed here, since it seems that it refers                                                                                                                                                                                                                                                                                                                                                                                                                                                                                                                                       | Change made.                                                                                                                                                                                                                                                                                                                                                            |

| to the bush and alley soils which are       |                                           |
|---------------------------------------------|-------------------------------------------|
| specified in their respective places in the |                                           |
| sentence. Check and reword for clarity.     |                                           |
| Line 267 - combine sentence with            | All similar paragraphs to that as         |
| previous paragraph. In fact, there are      | described have been adjusted throughout   |
| many spots like this throughout the text    | the text to better improve flow and place |
| where a sentence or two stand as their      | focus on the discussed topic, improving   |
| own paragraph (e.g., line 362, 393, 440,    | the conciseness, readability and          |
| 443, 447, 452, etc). This disrupts the flow | comprehension of the text.                |
| and connections between findings.           |                                           |
| Please find and rearrange as needed.        |                                           |